# Characterising a healthy adult with a rare HAO1 knockout to support a therapeutic strategy for primary hyperoxaluria

Tracy L McGregor[1], Karen A Hunt[2], Elaine Yee[1], Dan Mason[3], Paul Nioi[1], Simina Ticau[1], Marissa Pelosi[1], Perry R Loken[4], Sarah Finer[2], Deborah A Lawlor[5,6,7], Eric B Fauman[8], Qin Qin Huang[9], Christopher J Griffiths[2], Daniel G MacArthur[10,11], Richard C Trembath[12], Devin Oglesbee[4], John C Lieske[4], David V Erbe[1], John Wright[3], David A van Heel[2]*

[1]Alnylam Pharmaceuticals, Cambridge, United States; [2]Blizard Institute and Institute for Population Health Sciences, Barts and The London School of Medicine and Dentistry, Queen Mary University of London, London, United Kingdom; [3]Bradford Institute for Health Research, Bradford Teaching Hospitals National Health Service (NHS) Foundation Trust, Bradford, United Kingdom; [4]Mayo Clinic, Division of Nephrology and Hypertension, Rochester, United States; [5]MRC Integrative Epidemiology Unit at the University of Bristol, Oakfield House, Oakfield Grove, Bristol, United Kingdom; [6]Population Health Science, Bristol Medical School, Bristol University, Bristol, United Kingdom; [7]Bristol NIHR Biomedical Research Centre, Bristol, United Kingdom; [8]Internal Medicine Research Unit, Pfizer Worldwide Research, Development and Medical, Cambridge, United States; [9]Wellcome Sanger Institute, Hinxton, United Kingdom; [10]Analytic and Translational Genetics Unit, Massachusetts General Hospital, Boston, United States; [11]Program in Medical and Population Genetics, Broad Institute of MIT and Harvard, Cambridge, United States; [12]School of Basic and Medical Biosciences, Faculty of Life Sciences and Medicine, King's College London, London, United Kingdom

*For correspondence:
d.vanheel@qmul.ac.uk

**Abstract** By sequencing autozygous human populations, we identified a healthy adult woman with lifelong complete knockout of *HAO1* (expected ~1 in 30 million outbred people). *HAO1* (glycolate oxidase) silencing is the mechanism of lumasiran, an investigational RNA interference therapeutic for primary hyperoxaluria type 1. Her plasma glycolate levels were 12 times, and urinary glycolate 6 times, the upper limit of normal observed in healthy reference individuals (n = 67). Plasma metabolomics and lipidomics (1871 biochemicals) revealed 18 markedly elevated biochemicals (>5 sd outliers versus n = 25 controls) suggesting additional HAO1 effects. Comparison with lumasiran preclinical and clinical trial data suggested she has <2% residual glycolate oxidase activity. Cell line p.Leu333SerfsTer4 expression showed markedly reduced HAO1 protein levels and cellular protein mis-localisation. In this woman, lifelong *HAO1* knockout is safe and without clinical phenotype, de-risking a therapeutic approach and informing therapeutic mechanisms. Unlocking evidence from the diversity of human genetic variation can facilitate drug development.

## Introduction

We exome sequenced two large population-based cohorts with substantial autozygosity (Born In Bradford, Genes and Health [*Narasimhan et al., 2016*; *Finer, 2018*]). A healthy adult woman with

complete lifelong germline knockout of *HAO1* (hydroxyacid oxidase 1, encoding glycolate oxidase) was identified. The frameshift variant, rs1186715161 (present on a single Ensembl 99 transcript ENST00000378789.3:c.997delC, ENSP00000368066.3:p.Leu333SerfsTer4), for which the woman is homozygous, is predicted to truncate the final 37 amino acids of the HAO1 protein. Inspection of blood DNA exome sequencing read data confirmed the genotype (*Figure 1A*), as did Sanger dideoxy sequencing in a further saliva DNA sample (*Figure 1B*). The volunteer was 7.4% autozygous at the DNA level and the genotype was within a long (5.2 Mb) autozygous region. She was not homozygous for any other rare (minor allele frequency <1%) predicted loss-of-function genotypes.

No complete knockout (homozygous predicted loss of function, pLOF) variants in *HAO1* were observed in the online gnomAD v2.1 database of 141,456 whole exome/genome sequenced individuals (*Karczewski, 2020*; *Minikel et al., 2019*). The cumulative frequency of known pLOF variants in *HAO1* is low, in gnomAD v2.1 approximately 1 in 2700 individuals is heterozygous. Analysis of pLOF constraint, to assess the degree of natural selection against loss of function, revealed 13 variants observed out of 17.7 expected (obs/exp = 74%, 90% CI 48–117%), with a confidence interval spanning the range for recessive disease genes (mean obs/exp = 59%) and homozygous pLoF tolerant genes (mean obs/exp = 92%) (*Minikel et al., 2019*). The lack of strong constraint makes the *HAO1* gene unlikely to have a haploinsufficient phenotype, but does not rule out a deleterious health effect in homozygotes or a mild phenotype in heterozygous carriers. However, since the expected homozygote frequency in an outbred population would be only ~1 in 30 million individuals, assessing the health effects of *HAO1* loss to date has been challenging. Three children have previously been reported with suspected glycolate oxidase deficiency, however all were ascertained to have complicating factors such as multiple gene deletions or additional mutations associated with a severe genetic disease, thus confounding clinical interpretation. Furthermore none had plasma biochemical assays, and not all had *HAO1* sequencing performed (*Frishberg et al., 2014*), (*Clifford-Mobley et al., 2017*), (*Craigen, 1996*).

*HAO1* inhibition is a potential chronic therapeutic approach for a devastating metabolic disease (primary hyperoxaluria type 1, PH1). Thus, the phenotype of a *HAO1* null individual could illuminate the mechanism and safety profile of drugs targeting this enzyme (*Nguyen et al., 2018*). Her medical history, and UK National Health Service (NHS) primary and secondary care health records were reviewed. At the time of assessment, she was in her 5th decade and a mother with three healthy children. She declared British-Pakistani ethnicity. Renal ultrasound, carried out as part of a gynaecology assessment, was normal. She was overweight (BMI 30–35 kg/m$^2$), and other than common non-serious short-term illnesses and pregnancy symptoms, otherwise healthy.

Standard clinical venous blood biochemistries including serum sodium, potassium, bicarbonate, chloride, creatinine, transaminases, and bilirubin were repeatedly normal at recall and over the previous decade. At genotype directed recall, her serum anion gap was normal, and plasma and urinary oxalate were both normal. However, confirming the loss of function predicted from the genetics, her plasma glycolate and urinary glycolate levels were markedly elevated (*Table 1*), at, respectively, 12 times and six times the upper limit of normal in healthy reference individuals.

We analyzed 914 metabolites (736 known, 178 unknown, Metabolon HD4 Panel) and 957 lipids (Metabolon TrueMass Complex Lipid Panel) from her blood plasma and from controls. Consistent with the expectation that HAO1 plays a relatively limited metabolic role, the majority of her metabolites were similar (+/- 3 sd) compared to those of 25 control individuals. Principal component analysis of the metabolomics data showed her to be an outlier on PC2 (*Figure 2*), analysis of loadings suggested the high glycolate level to be the major driver. We observed that 18 biochemicals (including glycolate) were markedly elevated (extreme outliers at >5 sd compared to controls) in addition to other more modestly elevated biochemicals (*Supplementary file 1* and *2*). Some of these may reflect the known roles of glycolate in human metabolism, and be directly related to the very high glycolate. The elevated cholic acid related compounds might be due to reduced local availability of glycine (a product of the glycolate pathway) to conjugate cholic acid and its bile acid derivatives (unconjugated bile acids are elevated; glycine and taurine conjugated bile acids are reduced). Six of the significantly elevated metabolites have a structure compatible with being potential HAO1 substrates, two further evidenced by reduced levels of their predicted metabolic products (annotated in *Supplementary file 1*).

Whilst a liver biopsy with direct measurement of enzyme activity was not performed (because of the risk), we estimated activity from the relationship of *HAO1* silencing to substrate build up in vivo

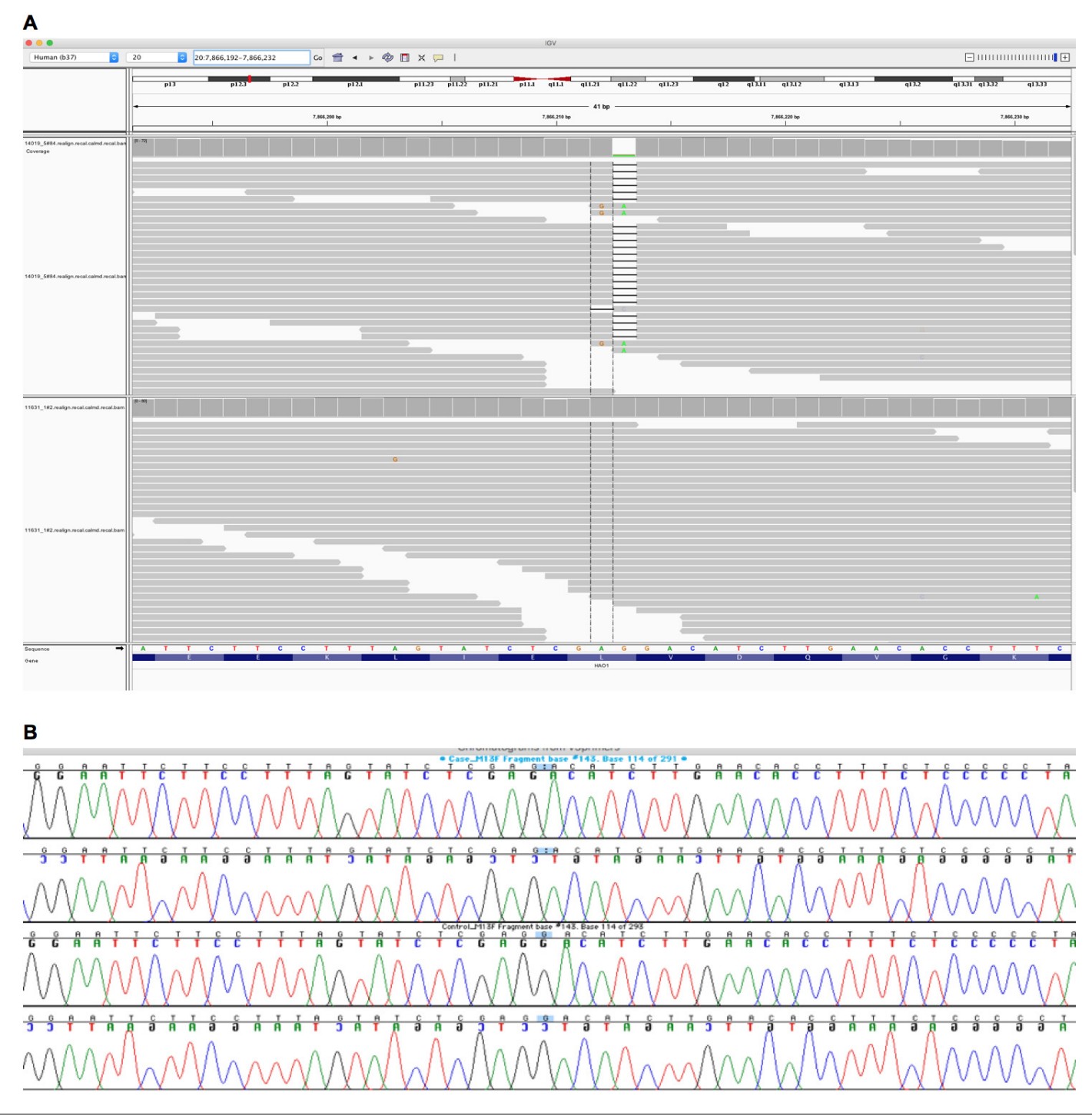

**Figure 1.** Integrative Genomics Viewer (IGV) image of sequencing alignments around the ENST00000378789.3:c.997delC *HAO1* variant (rs1186715161, GRCh37:20;7866212 AG/A). (**A**) Exome sequencing performed using Agilent in solution capture and Illumina short read sequencing using blood DNA. Top half of image, individual with homozygous ENST00000378789.3:c.997delC genotype. Bottom half of image, a different individual with homozygous reference sequence genotype. Neither sequencing assay (see also *Figure 1B*) in the different tissues suggested mosaicism however additional tissues (e.g. skin) for more definitive exclusion could not be obtained. (**B**) Sanger sequencing around the ENST00000378789.3:c.997delC *HAO1* (rs1186715161) variant using saliva DNA taken at a different timepoint to the blood DNA in *Figure 1A*.

**Table 1.** Blood and Urine biochemical measurements in a healthy woman with a p.Leu333SerfsTer4 *HAO1* knockout.

| Sample and assay | Level | Reference Range[*] |
|---|---|---|
| *Venous blood* | | |
| Plasma oxalate | <1.0 mcmol/L | <1.6 |
| Plasma glycolate | 171 nmol/mL | ≤14 |
| Plasma glycerate | <1 nmol/mL | ≤28 |
| Urine (single void, same day as blood tests) | | |
| Urine oxalate | 0.16 mmol/L | Not applicable |
| Urine oxalate/creatinine ratio | 20 mg/g | ≤75 |
| Urine glycolate/creatinine ratio | 199 mg/g | ≤50 |
| 24 hr urine (on a different day) | | |
| Urine oxalate (total/24 hr) | 0.27 mmol/24 hr | <0.46 |
| Urine oxalate/creatinine ratio | 23 mg/g | ≤75 |
| Urine glycolate/creatinine ratio | 309 mg/g | ≤50 |

[*]Reference ranges (mean+2sd) derived from studies of 67 healthy adults at the Mayo Clinic.

(*Liebow et al., 2017*; *Milliner Dawn, 2018*) using data from healthy volunteers in a phase one trial (32 individuals, https://clinicaltrials.gov/ct2/show/NCT02706886) with the investigational RNAi therapeutic lumasiran (Alnylam Pharmaceuticals) targeting *HAO1*. In healthy volunteers at the highest dose tested (6 mg/Kg) maximal plasma glycolate and urinary glycolate concentrations were 3–5 times less than observed in the *HAO1* knockout woman. This lumasiran dose would be expected to silence the *HAO1* mRNA at >95% based upon modeling from preclinical results in mice, rats, and monkeys and from direct comparison to clinical data from RNAi therapeutics to other targets that employ the same molecular design and have similar potency (*Fitzgerald et al., 2017*; *Hill et al., 2016*). Consequently, we estimate that this woman likely retains <2% residual glycolate oxidase activity.

HAO1 protein is normally found exclusively in hepatocyte peroxisomes and requires the C-terminal tripeptide SKI for this targeting (*Recalcati et al., 2001*). Since this woman's HAO1 protein is predicted to lack the entire C-terminus, we hypothesized an error in peroxisomal targeting might contribute to loss of function. We therefore performed in vitro cell line assays, transfecting plasmids with wild-type *HAO1* or encoding p.Leu333SerfsTer4 *HAO1* into cells (*Figure 3*). Results with p.Leu333SerfsTer4 revealed markedly reduced HAO1 protein (mRNA levels were unaffected) with the p.Leu333SerfsTer4 HAO1 protein also diffusely distributed (mis-localised) within cells.

PH1 is a rare autosomal recessive metabolic disorder of oxalate metabolism (*Cochat and Rumsby, 2013*). Overproduction of oxalate occurs in the liver due to an inherited genetic defect of the enzyme alanine-glyoxylate aminotransferase (encoded by *AGXT*). High concentrations of urinary oxalate result in precipitation of insoluble calcium oxalate salts and may lead to paediatric end stage kidney disease. Multi-organ damage can also occur from systemic oxalosis. Substantial unmet need exists for therapies to treat PH1 without requiring dual liver-kidney transplantation.

Targeting *HAO1* to reduce substrate for oxalate production in the liver is one potential therapeutic approach. Glycolate oxidase is directly upstream of AGXT, reducing glycolate oxidase activity is expected to decrease oxalate production and increase glycolate levels. Excess glycolate is expected to be freely excreted by the kidneys and is highly soluble. However, suppression of *HAO1* is a novel therapeutic approach that poses unknown risks. For example, as well as its glycolate oxidase activity, HAO1 can also metabolise 2-hydroxy fatty acids. Pre-clinical lumasiran studies in mice, rats, and monkeys suggested no adverse signals from chronic *HAO1* silencing of >95%, with the expected substantial elevations in glycolate levels (*Liebow et al., 2017*; *Martin-Higueras et al., 2016*; *Dutta et al., 2016*). Neither studies in non-human species nor relatively short-term therapeutic studies in people, fully inform the lifelong safety of glycolate oxidase targeting in humans, which remains unknown.

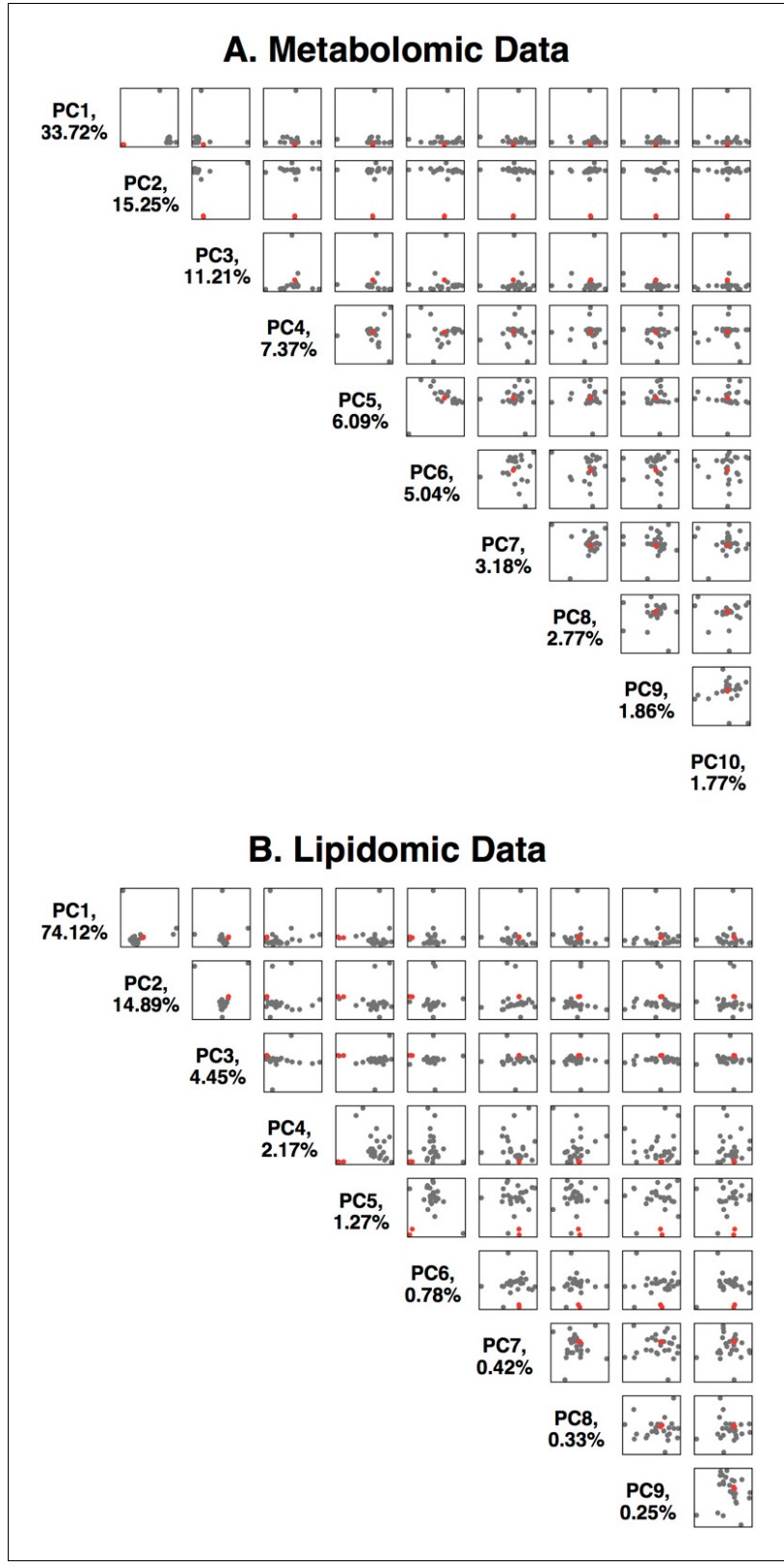

**Figure 2.** Principal component analysis of the Metabolon metabolomics data. (**A**) Data is available in *Supplementary file 1*. The first 10 principal components are plotted against each other, with %variance explained. The two replicates of the HAO1 knockout volunteer are shown as red dots, control individuals as black dots. (**B**) Principal component analysis of the Metabolon lipidomics data. Data is available in *Supplementary file 2*.

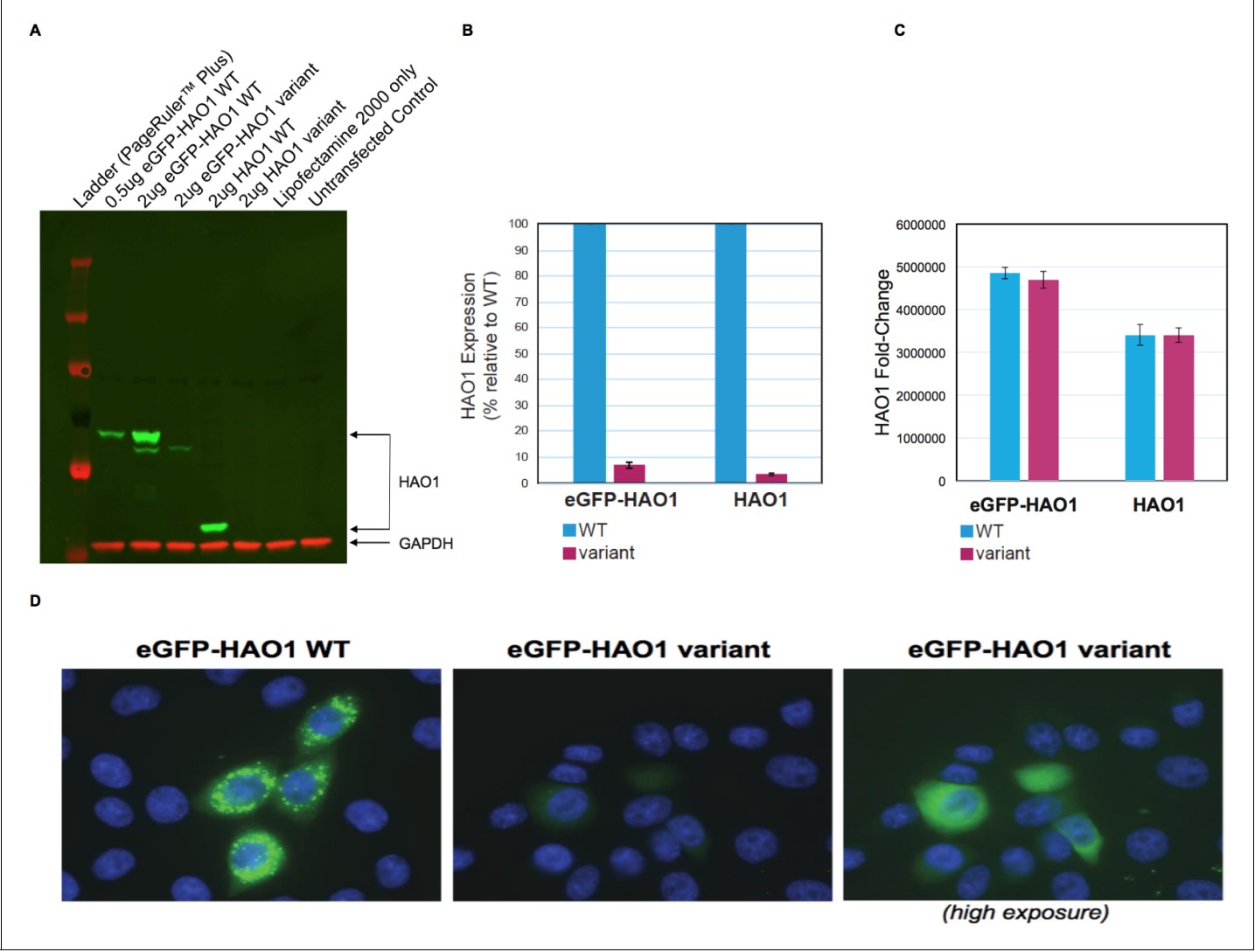

**Figure 3.** The *HAO1* variant p.Leu333SerfsTer4 is expressed at significantly lower levels and mislocalised in cells. (**A**) Western blot showing glycolate oxidase (GO, encoded by the *HAO1* gene) protein for both tagged and untagged reference sequence *HAO1* (wild type, WT) and p.Leu333SerfsTer4. (**B**) Protein expression quantification of both tagged and untagged wild type and p.Leu333SerfsTer4. (**C**) mRNA expression quantification of both tagged and untagged wild type and p.Leu333SerfsTer4. (**D**) Immunofluorescence shows lower expression levels and diffuse signal for p.Leu333SerfsTer4 relative to wild type. Cells imaged at 60X magnification. The right panel was adjusted to be 2.3-fold brighter than the left or middle panel in ImageJ software in order show green fluorescence.

A powerful approach to drug target de-risking is the identification of healthy people who naturally lack activity for a given protein (human knockouts). *PCSK9, LPA* and *APOC3* are classic examples (*Zhao et al., 2006*), (*Lim et al., 2014*), (*Saleheen et al., 2017*). Such reverse genetics - identifying individuals with extreme genotypes and then investigating their biology, provides an efficient alternative to the traditional approach of forward genetics - aggregating phenotypes and then investigating their genetic basis. A major challenge is that individuals with homozygous or compound heterozygous deficiency are expected to be extremely rare, made even more challenging to ascertain when the interest is in clinically silent phenotypes (e.g. to de-risk therapeutic targets). However, in selected populations where consanguineous unions are more common, the odds of homozygous mutations of all kinds increases profoundly due to autozygosity. Such naturally occurring genetic variation in autozygous humans provides opportunities to study the mechanism and safety of germline gene knockdown across the full life course of an individual - especially as studies are now sequencing adult populations, rather than focussing on children with rare diseases

(*Narasimhan et al., 2016*; *Saleheen et al., 2017*; *Mullard, 2017*). Our population genomic program, combined with excellent NHS e-health records; and the ability to directly recall consented subjects for further clinical research (*Corbin et al., 2018*) has allowed us to identify a clinically silent (at least in her genetic background, note a possible limitation of this study is that the effect of the knockout on other human genetic backgrounds could not be evaluated), but very informative, genotype leading to *HAO1* deficiency (and previously *PRDM9* deficiency in another individual *Narasimhan et al., 2016*). The study of even a single human is equivalent to a typical knockout mouse study, in that such mice are very highly inbred and near-identical, thus using multiple mice only increases measurement precision not the assessment of multiple genetic backgrounds. We propose that an international collaborative database of human knockouts could increase understanding of human physiology and accelerate development of therapies for a broad range of rare and common diseases. Such a database would not require a clinical phenotype for enrollment, and might identify most homozygous pLOF variants that are compatible with life. Different genes would be expected to yield different numbers of homozygous knockouts but even an n of 1, as reported here, provides profound insights.

Most importantly, our data support the potential safety of *HAO1* inhibition as a chronic therapy for the devastating metabolic disease PH1. Furthermore, any potential other effects of *HAO1* suppression beyond glycolate oxidation did not lead to any apparent clinical phenotype. This case demonstrates the value of studying the lifelong complete knockdown of a target protein in a living human to aid development of a potential therapeutic, both in de-risking the approach and providing in vivo mechanistic understanding to optimize its development. Furthermore, lifelong therapy for PH1 is likely to be required. Our approach demonstrates the potential for improved drug discovery through unlocking relevant evidence hiding in the diversity of human genetic variation.

The top two traces (forward sequencing, then below, reverse sequencing) are of the woman with homozygous ENST00000378789.3:c.997delC genotype. Bottom two traces a different individual with homozygous reference sequence genotype.

## Materials and methods

**Key resources table**

| Reagent type (species) or resource | Designation | Source or reference | Identifiers | Additional information |
|---|---|---|---|---|
| Cell line (*Chinese Hamster Ovary*) | CHO-K1 | ATCC CCL-61 | | RRID:CVCL_0214 |
| Commercial assay or kit | Taqman Assay for RT-qPCR HAO1 FAM | Thermofisher | Assay Id Hs00213909_m1 | |
| Commercial assay or kit | Taqman Assay for RT-qPCR GAPDH VIC | Thermofisher | Assay Id Cg04424038_gH | |

### Ethics

The *HAO1* knockout volunteer took part in both the Born In Bradford study and the Genes and Health study. Volunteers providing control samples took part in the Genes and Health study. Ethical approval was obtained from Bradford National Research Ethics Committee (06/Q1202/48) and the South East London National Research Ethics Committee (14/LO/1240). Informed consent and consent to publish was obtained. Information Sheets and Consent Forms are available on the study website (http://www.genesandhealth.org/volunteer-information).

### Exome sequencing

Exome sequencing was performed using Agilent in solution capture and Illumina short read sequencing as described in detail in *Narasimhan et al. (2016)*. This experiment was performed once. Data reported in the paper are available at the European Genotypephenome Archive (www.ebi.ac.uk/ega) under accession numbers EGAS00001000462, EGAS00001000511, EGAS00001000567, EGAS00001000717 and EGAS00001001301.

## Sanger sequencing

Sanger sequencing was performed as described in detail in *Narasimhan et al. (2016)*. Samples were amplified using M13 tagged primer pair below (PCR conditions: Initial denaturation: 10 min at 96℃; followed by 35 cycles of 15 s at 95℃; 15 s at 55℃; 30 s at 72℃ with a final extension at 72℃ for 5mins) before sending for PCR clean up and sequencing with M13F and M13R in-house primers at Source Bioscience. Forward: 5'TGTAAAACGACGGCCAGTTCAAATTCACTTCTCTCCACCA, Reverse: 5'CAGGAAACAGCTATGACCTGGGGCTTAGCTTTCCAGG. This experiment was performed once, but replicated in forward versus reverse, and versus the exome sequencing and other results..

## Variant annotation

The HAO1 variant rs1186715161 is present on a single transcript in Ensembl 99 (which uses GEN-ODE v33 BASIC); and on a single transcript in each of the GENCODE v32 comprehensive, NCBI RefSeq, CCDS (consensus coding sequence) annotations in the USCS genome browser. Whilst additional transcripts are possibly suggested by more experimental tools such as vizER (using GTEx RNA-seq data, snca.atica.um.es/browser) or older tools such as AceView (ncbi.nlm.nih.gov/IEB/Research/Acembly/) our in vivo biological data in the human volunteer with this homozygous variant supports complete functional knockout.

## Plasma and urine collection

Non-fasting blood samples were collected from the *HAO1* knockout individual and a control (control data not shown) for the bespoke biochemical assays. Samples sent for testing at the Mayo Clinical Laboratories were prepared as per specimen collection guidelines stipulated for each test by the Mayo Clinical Laboratories. Specifically 4 × 6.0 ml Na Hep vacutainers were filled with blood followed by inversion 10 times. Samples were centrifuged (1500xg, 10 min, slow deceleration) at 4℃, within 45 min of collection. 3 × 1.0 ml aliquots of plasma were prepared in 2.0 ml cryovials which were frozen on dry ice before storing at −80℃. A further 2.0 ml aliquot of plasma was aliquoted into a 3.0 ml cryovial which was then acidified to 2.3 < pH < 3.5 using concentrated (12N) hydrochloric acid, samples were then frozen on dry ice before storing at −80℃.

Urine samples were collected either at the visit, or over a 24 hr period, in both cases, the total volume was noted, collected sample was swirled to mix then 8–10 ml aliquots were prepared in 15 ml falcons. Samples were frozen on dry ice before storing at −80℃.

For the bespoke assays at the Mayo, one control sample was sent along with the *HAO1* knockout woman, and Mayo Clinical Labs were blinded to sample identity. Control samples for reference ranges at Mayo Clinical Labs were other samples being processed at Mayo at similar times. Assays at the Mayo were performed once.

Metabolon assays were carried out on plasma obtained from the *HAO1* knockout woman along with 25 controls. Control samples were of adults of south Asian ethnicity and both genders from the Genes and Health study. Demographics are provided in *Supplementary file 1*. For each person, a 6.0 ml EDTA vacutainer was filled with blood, vacutainers were then inverted five times before centrifugation (2200xg, 15mins, slow deceleration) at room temperature. 0.25 ml plasma was aliquoted into 2.0 ml pre-chilled cryovials, on ice, before immediate transfer to −80℃ freezer for storage until samples were transferred to Metabolon on dry ice. Metabolon assays were performed in duplicate (technical replicate) in the HAO1 knockout individual.

## Bespoke biochemical assays

Mayo Clinical Labs were blinded to the identity of the *HAO1* knockout individual, and a control individual. Urine and plasma oxalate and glycolate quantification was performed at the Mayo Clinical Labs with samples prepared as instructed for each test (see https://www.mayocliniclabs.com/test-catalog). Urine oxalate (Test ID: ROXU) was measured using a modification of the oxalate oxidase method as previously described (*Wilson and Liedtke, 1991*). Plasma oxalate (Test ID: POXA) concentrations were determined for an acidified sample (to minimize spontaneous conversion of ascorbate to oxalate) by ion-exchange chromatography using the Dionex ICS 2100 instrument. Urine glycolate (Test ID: HYOX) was determined along with a panel of metabolites (4-hydroxy-2-oxoglutaric, HOG; oxalate, glyoxylate, and glycerate). Briefly, urine samples corresponding to 0.25 mg of

creatinine (not to exceed 1 mL of urine) were oximated by reaction with methoxyamine hydrochloride to stabilize one of the target analytes (HOG). The urine was then acidified and extracted with 4:1 ethyl acetate:isopropanol. After evaporation, the dry residue was silylated with 80:20 BSTFA/1% TMCS:pyridine and analyzed by capillary gas chromatography/mass spectrometry (GC/MS) for quantification of each analyte. For plasma glycolate quantification (Test ID: HYOXP), non-acidified plasma samples were spiked with a mixture of stable isotope internal standards and treated with pentafluorobenzyl hydroxylamine HCl to prepare PFB-oxime derivatives of any oxo-acids present. After oximation, available hydroxy groups (alcohols or carboxylic acids) were derivatized with a silylating reagent (N,O-Bis(trimethylsilyl)trifluoroacetamide; BSTFA) to impart necessary volatility and stability for analysis by capillary GC-MS in PCI SIM mode. Quantification was enabled by calibration with use of internal standard, and one quantifier ion and one qualifier ion.

## Metabolon metabolomics (UPLC-MS/MS), Metabolon HD4 panel

Following receipt by Metabolon, samples were inventoried and immediately stored at −80℃. Each sample received was accessioned into the Metabolon LIMS system and was assigned by the LIMS a unique identifier that was associated with the original source identifier only. This identifier was used to track all sample handling, tasks, results, etc. The samples (and all derived aliquots) were tracked by the LIMS system. All portions of any sample were automatically assigned their own unique identifiers by the LIMS when a new task was created; the relationship of these samples was also tracked. All samples were maintained at −80℃ until processed.

Samples were prepared using the automated MicroLab STAR system from Hamilton Company. Several recovery standards were added prior to the first step in the extraction process for QC purposes. To remove protein, dissociate small molecules bound to protein or trapped in the precipitated protein matrix, and to recover chemically diverse metabolites, proteins were precipitated with methanol under vigorous shaking for 2 min (Glen Mills GenoGrinder 2000) followed by centrifugation. The resulting extract was divided into five fractions: two for analysis by two separate reverse phase (RP)/UPLC-MS/MS methods with positive ion mode electrospray ionization (ESI), one for analysis by RP/UPLC-MS/MS with negative ion mode ESI, one for analysis by HILIC/UPLC-MS/MS with negative ion mode ESI, and one sample was reserved for backup. Samples were placed briefly on a TurboVap (Zymark) to remove the organic solvent. The sample extracts were stored overnight under nitrogen before preparation for analysis.

Several types of controls were analyzed in concert with the experimental samples: a pooled matrix sample generated by taking a small volume of each experimental sample (or alternatively, use of a pool of well-characterized human plasma) served as a technical replicate throughout the data set; extracted water samples served as process blanks; and a cocktail of QC standards that were carefully chosen not to interfere with the measurement of endogenous compounds were spiked into every analyzed sample, allowed instrument performance monitoring and aided chromatographic alignment. Instrument variability was determined by calculating the median relative standard deviation (RSD) for the standards that were added to each sample prior to injection into the mass spectrometers. Overall process variability was determined by calculating the median RSD for all endogenous metabolites (i.e. non-instrument standards) present in 100% of the pooled matrix samples. Experimental samples were randomized across the platform run with QC samples spaced evenly among the injections.

The HD4 dataset comprises a total of 913 biochemicals, 735 compounds of known identity (named biochemicals) and 178 compounds of unknown structural identity (unnamed biochemicals). Analysis of glycolate (normally undetectable in this assay) was specifically added by Metabolon as an extra measurement to the standard 913 panel.

We used the ScaledImpData results set provided by Metabolon. Here, each biochemical in the OrigScale results set (values normalized in terms of raw area counts) is rescaled to set the median equal to 1. Then, below detection limit (missing) values are imputed with the minimum.

Principal components analysis was performed for metabolomics (and lipidomics separately) using the 'PCAtools' R package (version 1.2.0). We used the scaled data prepared by Mebabolon which are available in the ScaledImpData worksheets of *Supplementary file 1*. Data were not further scaled since the extreme values observed are expected to be driven by true biological signals.

Excellent correlation (r = 0.9976) between two duplicates of the HAO1 knockout samples were observed. Therefore the mean of the two HAO1 knockout samples was compared to the mean of

the 25 other controls, and extreme differences reported (full individual level data for all biochemicals is provided in *Supplementary file 1*).

## Metabolon lipidomics (infusion MS), Metabolon CLP panel

The extracts were dried under nitrogen and reconstituted in a dichloromethane:methanol solution containing ammonium acetate. The extracts were transferred to vials for infusion-MS analysis, performed on a Shimadzu LC with nano PEEK tubing and the Sciex SelexIon-5500 QTRAP. The samples were analyzed via both positive and negative mode electrospray. The 5500 QTRAP was operated in MRM mode with a total of more than 1100 MRMs. Individual lipid species were quantified by taking the ratio of the signal intensity of each target compound to that of its assigned internal standard, then multiplying by the concentration of internal standard added to the sample. Lipid class concentrations were calculated from the sum of all molecular species within a class, and fatty acid compositions were determined by calculating the proportion of each class comprised by individual fatty acids.

Lipids were extracted from the biofluid in the presence of deuterated internal standards using an automated BUME extraction according to the method of Lofgren et al. (J Lipid Res 2012:53(8):1690–700).

We used the ScaledImpData results set provided by Metabolon. Here, each biochemical in the OrigScale results set (values normalized in terms of raw area counts) is rescaled to set the median equal to 1. Then, below detection limit (missing) values are imputed with the minimum.

Principal components analysis was performed for lipidomics using the 'PCAtools' R package (version 1.2.0). We used the scaled data prepared by Mebabolon which are available in the ScaledImp-Data worksheets of *Supplementary file 2*. Data were not further scaled since the extreme values observed are expected to be driven by true biological signals.

Excellent correlation (r = 0.9737) between two duplicates of the HAO1 knockout samples were observed. Therefore, the mean of the two HAO1 knockout samples was compared to the mean of the 25 other controls, and extreme differences reported (full individual level data for all biochemicals is provided in *Supplementary file 2*).

## Cell line expression assays

Plasmids: Plasmids were generated by GenScript and were verified using Sanger sequencing.

Cell culture and transfection: CHO-K1 (ATCC CCL-61) were cultured in F-12K Medium supplemented with 10% FBS. This cell line was authenticated (cytochrome C oxidase I gene (COI) analysis) and confirmed mycoplasma free by the American Type Culture Collection (ATCC). Cells were grown in tissue-culture treated plates. After 18 hr, half of the media was replaced with fresh media and the cells were transiently transfected with plasmid using Lipofectamine 2000, following manufacturer's instructions.

Immunofluorescence microscopy: 24 hr after transfection, CHO-K1 cells were re-plated into chamber slides. After 24 hr, cells were washed with PBS, fixed for 20 min with 4% paraformaldehyde, and washed with PBS for 5 min. The cells were permeabilized and blocked in PBS containing 1%BSA/9% normal goat serum/0.3M glycine/0.1% Tween for 1–3 hr. Cells were washed with PBS and coverslips were mounted on the slides with Antifade Mounting Medium with DAPI (VECTASHIELD; Cat. No: H-1200). Immunofluorescence was analyzed on Revolve microscope (ECHO) and images were processed using ImageJ software.

Protein Isolation: Cells were lysed in RIPA buffer (Thermo Scientific Catalog number: 89900) supplemented with 1X protease inhibitor (Roche; Catalog number: 4693132001) at 4°C for 15 min. Afterwards, the samples were centrifuged at 16,000 xg for 30 min at 4°C and the supernatants were collected for downstream processing.

Western Blot: Cell lysate samples were mixed in equal volume of Tris-Glycine SDS Sample Buffer (2X) with 5% BME (Invitrogen Novex; Catalog number: LC2676). Samples were boiled at 100°C for 5 min and cooled on ice. Protein ladder (Thermo Scientific; Catalog number: 26619) and samples were loaded into 8% Tris-Glycine Mini Gels, (Invitrogen Novex; Catalog number: XP00085BOX) in Mini Gel Tank (Life Technologies; Catalog number: A25977). Gel ran at 100V and was transferred to nitro-cellulose membrane with iBlot 2 Dry Blotting System (Invitrogen; Catalog Number: IB21001), following manufacturer's protocol. The membrane was blocked for 30 min with 1X TBS blocking buffer

(Thermo Scientific; Catalog number: 37537) and washed three times with 1X TBS Tween 20 Buffer (Thermo Scientific; Catalog number: 28360). Membrane was incubated in primary Anti-GO antibody (Abcam; ab93137) at 1:500 dilution and primary Anti-GAPDH antibody (Abcam; ab8245): at 1:2500 dilution overnight at 4°C. Next day, membrane was washed three times and incubated at room temperature in Goat anti-Rabbit IgG Secondary Antibody (Li-Cor; 926–32211) and Goat anti-Mouse IgG Secondary Antibody (Li-Cor; 926–68070) both at 1:7500 dilution. Membrane was washed three times and imaged using the Bio-Rad ChemiDoc MP Imaging System.

RNA isolation: RNA was extracted and DNase-treated using QIAgen RNeasy Mini Kit (Cat No./ID: 74104), following manufacturer's instructions.

RT-qPCR: RNA was reverse transcribed into cDNA using High-Capacity cDNA Reverse Transcription Kit (Applied Biosystems; PN 4375575), following manufacturer's protocol. qPCR was performed using Lightcycler 480 probe master mix (Cat # 04887301001) and probed with HAO1 FAM (Assay Id Hs00213909_m1, https://www.thermofisher.com/taqman-gene-expression/product/Hs00213909_m1?CID=&lCID=&subtype which spans exons 2 and 3) and control GAPDH VIC non-PL (Assay Id Cg04424038_gH) at 95°C for 10 min, amplified for 45 cycles at 95°C for 10 s and 60°C for 30 s and 72°C for 1 s, and cooled at 40°C for 30 s. Cell line assays were performed twice.

## Acknowledgements

Genes and Health is funded by Wellcome (WT102627, WT210561), the Medical Research Council (UK) (M009017), Higher Education Funding Council for England Catalyst, Barts Charity (845/1796), Health Data Research UK (for London substantive site), and research delivery support from the NHS National Institute for Health Research Clinical Research Network (North Thames).

Born In Bradford is funded by Wellcome (WT101597MA), a joint grant from the UK Medical Research Council (MRC) and UK Economic and Social Science Research Council (ESRC) (MR/N024397/1), the British Heart Foundation (CS/16/4/32482) and the National Institute for Health Research (NIHR) under its Collaboration for Applied Health Research and Care (CLAHRC) for Yorkshire and Humber. DAL is supported by, the British Heart Foundation (AA/18/7/34219), US National Institute of Health (R01 DK10324) and the European Research Council (669545); she works in a Unit that is supported by the University of Bristol and the MRC (MC_UU_00011/6), which supported DNA extraction from participants in BiB.

Genes and Health received an unrestricted grant from Alnylam Pharmaceuticals for work performed in this manuscript. Born In Bradford received funding from Alnylam Pharmaceuticals for data extraction from clinical records. Alnylam Pharmaceuticals directly funded biochemical assays performed at the Mayo Clinic.

We thank Eric Minikel for comments on gene constraint analyses. We thank Dr John Knight and Dr Hoy Shen for comments on the metabolomics data.

We thank all of the volunteers participating in Genes and Health and Born in Bradford, and the staff who have recruited and collected data from volunteers.

The views expressed in this paper are those of the authors and not necessarily any funders or others acknowledged here.

## Additional information

### Competing interests

Tracy L McGregor, Elaine Yee, Paul Nioi, Simina Ticau, Marissa Pelosi, David V Erbe: employee of Alnylam Pharmaceuticals. Eric B Fauman: is affiliated with Pfizer Worldwide Research. The author has no financial interests to declare. Contributed as an individual and the work was not part of a Pfizer collaboration nor was it funded by Pfizer. The other authors declare that no competing interests exist.

### Funding

| Funder | Grant reference number | Author |
| --- | --- | --- |
| Wellcome | WT102627 | David A van Heel |

| Wellcome | WT210561 | David A van Heel |
| Medical Research Council | M009017 | David A van Heel |

The funders had no role in study design, data collection and interpretation, or the decision to submit the work for publication.

## Author contributions

Tracy L McGregor, Conceptualization, Supervision, Funding acquisition, Project administration, Writing - review and editing; Karen A Hunt, Formal analysis, Supervision, Validation, Investigation, Visualization, Methodology, Project administration, Writing - review and editing; Elaine Yee, Resources, Formal analysis, Validation, Investigation, Visualization, Methodology, Writing - review and editing; Dan Mason, Resources, Formal analysis, Investigation, Writing - review and editing; Paul Nioi, Supervision, Validation, Methodology, Writing - review and editing; Simina Ticau, Marissa Pelosi, Investigation, Writing - review and editing; Perry R Loken, Devin Oglesbee, John C Lieske, Investigation, Methodology, Writing - review and editing; Sarah Finer, Deborah A Lawlor, Christopher J Griffiths, Resources, Writing - review and editing; Eric B Fauman, Methodology, Writing - review and editing; Qin Qin Huang, Formal analysis; Daniel G MacArthur, Resources, Formal analysis, Investigation, Methodology, Writing - original draft, Writing - review and editing; Richard C Trembath, Resources, Funding acquisition, Investigation, Methodology, Writing - original draft, Project administration, Writing - review and editing; David V Erbe, Conceptualization, Formal analysis, Funding acquisition, Investigation, Methodology, Writing - original draft, Project administration, Writing - review and editing; John Wright, Resources, Writing - original draft, Project administration, Writing - review and editing; David A van Heel, Conceptualization, Resources, Data curation, Formal analysis, Supervision, Funding acquisition, Validation, Investigation, Visualization, Methodology, Writing - original draft, Project administration, Writing - review and editing

## Author ORCIDs

Sarah Finer http://orcid.org/0000-0002-2684-4653
John C Lieske http://orcid.org/0000-0002-0202-5944
David A van Heel https://orcid.org/0000-0002-0637-2265

## Ethics

Human subjects: The HAO1 knockout volunteer took part in both the Born In Bradford study and the Genes & Health study. Volunteers providing control samples took part in the Genes & Health study. Ethical approval was obtained from Bradford National Research Ethics Committee (06/Q1202/48) and the South East London National Research Ethics Committee (14/LO/1240). Informed consent and consent to publish was obtained.

## Decision letter and Author response

Decision letter https://doi.org/10.7554/eLife.54363.sa1
Author response https://doi.org/10.7554/eLife.54363.sa2

# Additional files

## Supplementary files

- Supplementary file 1. Metabolon HD4 metabolomic data.
- Supplementary file 2. Metabolon CLP lipidomic data.
- Transparent reporting form

## Data availability

All metabolomic data is available in full in the Supplementary Files.

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
