## [Decision Letter]

**Acceptance summary:**

Your comparison of the plasma metabolic and lipidomic profiles of a healthy adult with a life-long loss of function of the HAO gene provides support for the therapeutic approach of reducing HAO function in primary hyperoxaluria type 1. Furthermore the study demonstrates the value of identifying rare loss of function alleles in the healthy population to provide support for therapeutic strategies for the treatment of metabolic disorders.

**Decision letter after peer review:**

Thank you for submitting your article "Characterising a healthy adult with a rare HAO1 knockout to support a therapeutic strategy for primary hyperoxaluria." for consideration by *eLife*. Your article has been reviewed by three peer reviewers, and the evaluation has been overseen by a Reviewing Editor and Detlef Weigel as the Senior Editor. The following individuals involved in review of your submission have agreed to reveal their identity: Mina Ryten (Reviewer #2); Nicholas Blackburn (Reviewer #3).

The reviewers have discussed the reviews with one another and the Reviewing Editor has drafted this decision to help you prepare a revised submission.

Essential revisions:

Three reviewers have now assessed your manuscript and all agree that the data are both interesting and important. Although the reviewers agree that the paper is highly significant for the field, the metabolic consequences of the HAO1 deficit are rather descriptive in their current format. These data should be revised to better clarify the observed changes in metabolites particularly as related to lipid and amino acid metabolism. One approach would be to use principal component analysis of the combined Metabolon measurements to demonstrate how metabolically remarkable the HAO1 knockout individual is at a global metabolic level versus the 27 controls as well as with a focus on lipid and amino acid metabolism. These data should be included as a figure in the main text not as supplementary material in the final manuscript. The reviewers also request that the raw measurements are provided for the 1871 lipids and metabolites for the patient and the 27 control individuals, as well as basic demographic and anthropometric information (age, sex, height, weight). This is important given the small sample size of the overall study to assess potential biases.

A second issue raised by the reviewers is annotation of the variant, rs1186715161. Currently the HAO1 gene is reported by Ensembl (release 99) to generate a single transcript resulting in a single prediction of the impact of the c.997delC variant detected. However, given that HAO1 contains 8 exons and the vast majority of multi-exon genes are alternatively spliced, it is possible that alternative transcript structures exist which could change the interpretation of the variant identified. This view is supported by vizER, which uses RNAseq data to identify potential novel transcripts of known genes (https://www.biorxiv.org/content/10.1101/499103v2) and suggests the existence of novel transcripts of HAO1 in liver. With this in mind, the reviewers note that AceView predicts truncated transcripts of the gene. We therefore suggest that the authors review and include a discussion of other annotations and whether the interpretation of the variant remains stable across different annotation providers.

The reviewers also suggest that some caveats be included in the Discussion. While saliva and blood samples were used to test for mosaicism it is well recognised that saliva samples contain large quantities of blood-derived leukocytes, therefore this is not an effective means of demonstrating the absence of mosaicism. If it is possible to obtain a skin biopsy this would be a better way to test for mosaicism. If this is not possible within two months, please include a caveat to address this. Second the fact that the homozygous LoF mutation is studied only in one genetic background is a limitation of the analysis. We realize that examining the variant on multiple genetic backgrounds is beyond the scope of this work but please address this limitation in the Discussion.

---

## [Author Response]

Essential revisions:Three reviewers have now assessed your manuscript and all agree that the data are both interesting and important. Although the reviewers agree that the paper is highly significant for the field, the metabolic consequences of the HAO1 deficit are rather descriptive in their current format. These data should be revised to better clarify the observed changes in metabolites particularly as related to lipid and amino acid metabolism. One approach would be to use principal component analysis of the combined Metabolon measurements to demonstrate how metabolically remarkable the HAO1 knockout individual is at a global metabolic level versus the 27 controls as well as with a focus on lipid and amino acid metabolism. These data should be included as a figure in the main text not as supplementary material in the final manuscript.

We thank the reviewers for this useful suggestion. We performed the principal components analyses suggested, and show the HAO1 individual versus the controls. We provide these data as a main text figure. Dr Qin Qin Huang performed this analysis, and is added as a co-author.

We have preferred to treat the metabolomic and lipidomic data separately. This is because they are distinctly separate assays with completely different laboratory preparation and assay steps, and we believe this the correct way to analyse these data. We do now provide the full individual level dataset, should other journal readers wish to analyse the data in their own way.

Whilst the HAO1 individual is clearly identifiable in the 2^nd^ principal component of the metabolomics, this remains consistent with our existing statement in the text ‘Consistent with the expectation that HAO1 plays a relatively limited metabolic role, the vast majority of her metabolites were similar to those of 25 control individuals’.

The reviewers also request that the raw measurements are provided for the 1871 lipids and metabolites for the patient and the 27 control individuals, …

We thank the reviewer, and agree this is best practice for a scientific publication.

The raw measurements for all individuals are now added to both Supplementary file 1A and Supplementary file 1B as additional worksheets. We now mention in the Materials and methods text that these data are available.

As well as basic demographic and anthropometric information (age, sex, height, weight).

We have provided the information we have (age, sex, ethnicity) in the individual level data worksheet in Supplementary file 1A. We have partly anonymized the age to protect volunteer identity, by providing age range. We now mention in the Materials and methods text that these data are available. Unfortunately, we have incomplete data for body mass index (or height, weight) and therefore have not provided this.

Whilst providing this data, two individuals who were not of British South Asian ethnicity (as had been stated) were identified, and have now been removed from the controls. There are therefore now 25 controls. The analysis has been redone. The results, and their interpretation are essentially unchanged.

This is important given the small sample size of the overall study to assess potential biases.

We thank the reviewer. We have now provided as much of the requested data as we can. However we would remind the reviewer that the comparison is of 1 individual versus the 25 controls, and that we are reporting very gross changes (extreme outliers at >5sd) not small differences that might be subject to more subtle bias. We do now provide the full dataset such that readers can perform their own analyses should they wish.

A second issue raised by the reviewers is annotation of the variant, rs1186715161. Currently the HAO1 gene is reported by Ensembl (release 99) to generate a single transcript resulting in a single prediction of the impact of the c.997delC variant detected. However, given that HAO1 contains 8 exons and the vast majority of multi-exon genes are alternatively spliced, it is possible that alternative transcript structures exist which could change the interpretation of the variant identified. This view is supported by vizER, which uses RNAseq data to identify potential novel transcripts of known genes (https://www.biorxiv.org/content/10.1101/499103v2) and suggests the existence of novel transcripts of HAO1 in liver. With this in mind, the reviewers note that AceView predicts truncated transcripts of the gene. We therefore suggest that the authors review and include a discussion of other annotations and whether the interpretation of the variant remains stable across different annotation providers.

We thank Dr Ryten for these suggestions. We have added more Ensembl discussion to the main text (first paragraph). We have added to the Materials and methods a new section called ‘Variant Annotation’ including discussion of commonly used annotations (GENCODE comprehensive and basic, RefSeq, CCDS, etc.) in both Ensembl and UCSC, and the more experimental vizER and AceView annotations.

The reviewers also suggest that some caveats be included in the Discussion. While saliva and blood samples were used to test for mosaicism it is well recognised that saliva samples contain large quantities of blood-derived leukocytes, therefore this is not an effective means of demonstrating the absence of mosaicism. If it is possible to obtain a skin biopsy this would be a better way to test for mosaicism. If this is not possible within two months, please include a caveat to address this.

Unfortunately the volunteer did not consent to a skin biopsy. We have added a caveat to this part of the Discussion (in Figure 1A legend) as requested. We note also that the plasma glycolate and other data do suggest a true functional knockout in vivo.

Second the fact that the homozygous LoF mutation is studied only in one genetic background is a limitation of the analysis. We realize that examining the variant on multiple genetic backgrounds is beyond the scope of this work but please address this limitation in the Discussion.

Yes we completely agree with the reviewer and this is an excellent point. We did have a little on this in the penultimate paragraph, and have now expanded this bit of the Discussion as the reviewer requests.